# Pleiotropic Effects of Acetylsalicylic Acid after Coronary Artery Bypass Grafting—Beyond Platelet Inhibition

**DOI:** 10.3390/jcm10112317

**Published:** 2021-05-26

**Authors:** Dominika Siwik, Magdalena Gajewska, Katarzyna Karoń, Kinga Pluta, Mateusz Wondołkowski, Radosław Wilimski, Łukasz Szarpak, Krzysztof J. Filipiak, Aleksandra Gąsecka

**Affiliations:** 11st Chair and Department of Cardiology, Medical University of Warsaw, Banacha 1a, 02-097 Warsaw, Poland; dominika.siwik@gmail.com (D.S.); gmgajewska@gmail.com (M.G.); katarzkar@gmail.com (K.K.); plutakinga.01@gmail.com (K.P.); krzysztof.filipiak@wum.edu.pl (K.J.F.); 2Department of Cardiac Surgery, Medical University of Warsaw, 02-097 Warsaw, Poland; mateusz.wondolkowski@gmail.com (M.W.); rwilimski@gmail.com (R.W.); 3Bialystok Oncology Center, 15-027 Bialystok, Poland; lukasz.szarpak@gmail.com; 4Maria Sklodowska-Curie Medical Academy in Warsaw, 00-001 Warsaw, Poland

**Keywords:** acetylsalicylic acid, ASA, CABG, coronary artery bypass grafting, Alzheimer’s disease, hypertension, osteoporosis, cancer, inflammation, atherosclerosis

## Abstract

Acetylsalicylic acid (ASA) is one of the most frequently used medications worldwide. Yet, the main indications for ASA are the atherosclerosis-based cardiovascular diseases, including coronary artery disease (CAD). Despite the increasing number of percutaneous procedures to treat CAD, coronary artery bypass grafting (CABG) remains the treatment of choice in patients with multivessel CAD and intermediate or high anatomical lesion complexity. Taking into account that CABG is a potent activator of inflammation, ASA is an important part in the postoperative therapy, not only due to ASA antiplatelet action, but also as an anti-inflammatory agent. Additional benefits of ASA after CABG include anticancerogenic, hypotensive, antiproliferative, anti-osteoporotic, and neuroprotective effects, which are especially important in patients after CABG, prone to hypertension, graft occlusion, atherosclerosis progression, and cognitive impairment. Here, we discuss the pleiotropic effects of ASA after CABG and provide insights into the mechanisms underlying the benefits of treatment with ASA, beyond platelet inhibition. Since some of ASA pleiotropic effects seem to increase the risk of bleeding, it could be considered a starting point to investigate whether the increase of the intensity of the treatment with ASA after CABG is beneficial for the CABG group of patients.

## 1. Introduction

Acetylsalicylic acid (ASA), commonly known as aspirin, is one of the most renowned drugs of all times. It was firstly introduced into clinical practice in 1899. Its centuries-long fame remains vivid thanks to its antipyretic, analgesic, and anti-inflammatory effects. Yet, the ubiquitous ASA is generally indicated in the secondary prevention of major cardiovascular and neurovascular events. However, more and more studies indicate the pleiotropy of this well-known anti-inflammatory agent.

ASA in a standard dose of 75–160 mg once daily is commonly used for secondary prevention of thromboembolic events after coronary artery bypass grafting (CABG) [1]. CABG is the treatment of choice in patients with stable multivessel coronary artery disease (CAD) or left main CAD with intermediate or high anatomical complexity of CAD [2]. During CABG, an artery or vein from the patient’s body is used to create a bypass for the coronary artery around the significant atherosclerotic lesions. The goal of CABG is to provide adequate blood flow to the ischemic myocardium and thus restore its viability and function [3]. Patients after CABG procedure may benefit from ASA not only due to its antiplatelet activity, but also due to other features, including anti-inflammatory, anticancerogenic, hypotensive, antithrombotic, anti-osteoporotic, and neuroprotective effects.

Taking into account that CABG is a potent activator of inflammation, ASA is an important part in postoperative therapy not only as an antiplatelet action, but also as an anti-inflammatory agent [4,5]. Furthermore, since CAD and some types of cancer have common risk factors—including diabetes, obesity, and smoking—the use of ASA as a complementary therapy in cancer patients might limit the side effects from chemotherapeutics [6,7,8]. Moreover, ASA seems to have a hypotensive effect, especially when administered in the evening with statins, which indicates the benefits of the combined therapy after CABG. ASA therapy also prevents excessive and dysfunctional vascular smooth muscle cell proliferation (VSMCs), thereby decreasing the risk of late graft failure and further atherosclerosis progression [9]. Furthermore, ASA may increase bone marrow density (BMD) and reduce the potential fracture risk after CABG, thus facilitating rehabilitation. Finally, the neuroprotective features of ASA might be especially beneficial in elderly patients after CABG, who are susceptible to cognitive impairment and dementia [10].

CABG can be performed with (on-pump) or without cardiopulmonary bypass application (off-pump). On- and off-pump CABG have a different effect on inflammatory response and platelet homeostasis [11,12]. Off-pump CABG affects platelet function less than the on-pump CABG and is associated with higher rate of high on-ASA platelet reactivity after the operation [13].

Though well-recognized for its beneficial effects, ASA is also associated with several serious complications, with increased risk of bleeding being the most important one. In 2018 Tsoi et al. compared clinical outcomes of 204,170 ASA users and 408,339 non-users and observed significantly higher incidence of gastrointestinal bleeding in long-term aspirin users (4.64% vs. 2.74%) [14]. A meta-analysis of 35 randomized controlled trials showed a 55% increased risk for a major gastrointestinal bleeding due to ASA administration in daily doses of 75 to 325 mg [15]. The risk of bleeding associated with ASA has been established to be dose-related and further elevated by combining it with other antiplatelet agents or gastrotoxic drugs such as corticosteroids [16].

ASA has been examined several times as a potential drug for primary prevention of cardiovascular diseases and most of the studies concluded that ASA offered little to no benefit for primary prevention, while it increased the risk of major bleeding [17]. Recently, a randomized controlled trial of healthy participants aged ≥70, who were randomly assigned to receive 100 mg of ASA or placebo, confirmed no significant difference in the risk of cardiovascular disease, but markedly higher risk of major hemorrhage was significantly higher in the ASA group, than in placebo group (8.6 events per 1000 vs. 6.2 events per 1000 person-years, respectively) [18].

The threat of bleeding is the most important concern when it comes to the use of ASA in cardiac surgery. Several randomized controlled trials conducted in the 1980s and 1990s showed a rise in transfusion, re-exploration, and chest tube drainage in patients treated with preoperative ASA [19,20,21,22]. However, the results of more recent studies do not entirely confirm the previous findings. A 2015 study examined the effect of preoperative ASA administration in patients undergoing CABG, valve or combined CABG/valve surgery [23]. Preoperative ASA was associated with an increased incidence of red blood cell transfusion. On the contrary, a 2005 retrospective cohort study showed lower postoperative mortality rate among patients who received preoperative aspirin without significant increases in hemorrhage and transfusion [24].

In this review, we report data on the studies regarding off-pump CABG procedures [13]. We discuss the plethora of the underestimated applications of ASA, which might underlie its possible additional benefits in patients after CABG, who are a unique group of multimorbid patients. The accompanying diseases in this patient group include arterial hypertension (84%), hypercholesterolemia (47%), type 2 diabetes mellitus (32%), lower limb atherosclerosis (28%), chronic kidney disease (9%), and chronic obstructive pulmonary disease (9%) [13]. In addition, these elderly patients are more prone to post-operative wound healing disturbances and bleeding complications, and are at high risk of age-associated disease including osteoporosis, cancer, and cognitive impairment [25,26]. Based on ASA pleiotropic effects, we suggest that it could be considered to study the intensification of the postoperative treatment with ASA after CABG, by increasing either the frequency of ASA administration (e.g., 75–81 mg twice daily) or by increasing a single dose (e.g., 150–162 mg once daily).

The possible targets for ASA pleiotropic effects in patients after CABG are shown in Figure 1. The summary of ASA pleiotropic effects discussed in this review are summarized in Figure 2.

## 2. Mechanism of Action

ASA belongs to the non-salicylate nonsteroidal anti-inflammatory drugs (NSAIDs), which mechanism of action is based on cyclooxygenase (COX) inhibition. COX plays the main role in the conversion of arachidonic acid, which is released from membrane phospholipids, into prostaglandin H2 (PGH2) [27]. PGH2 has two active sites: one for peroxidase and one for COX. The latter is the binding site for ASA and NSAIDs. Eventually, COX by binding to the COX site of PGH2 leads to transformation into various prostaglandins (PGI2, PGE2, PGF2, PGD2) and thromboxane A2. By inhibiting COX, ASA reduces the inflammatory response and inhibits platelet function. The antiplatelet effect of ASA, in turn, results from inhibition of TXA2 production, which is formed in platelets from arachidonate by the aspirin-dependent COX pathway [28]. Since platelets are anucleated bone marrow fragments, unable to produce the COX-1 protein, the effect of ASA lasts for the lifetime of a platelet, around 10 days [29]. At least two different isoenzymes are the targets of ASA: COX-1 and COX-2. COX-1 is constitutively expressed on most cells, especially platelets and gastric mucosal cells. By stimulating prostaglandin production, COX-1 takes maintains homeostasis, for example activates platelets when required and protects the gastric mucosa. Conversely, COX-2 is periodically produced by the inflammatory cells in response to cytokine and growth factor stimulation. ASA irreversibly blocks both COX isoenzymes, but it has a stronger effect on COX-1, in comparison to COX-2. The dose-dependence of the ASA effect is crucial to understand its mode of action. Low doses of ASA (75–81 mg per day) inhibit COX-1, providing an antithrombotic effect. Intermediate doses (650 mg to 4 g per day) target both COX-1 and COX-2, exerting analgesic and antipyretic effects by impeding prostaglandin production. Finally, high doses (4–8 g per day) have anti-inflammatory effects [27].

## 3. Anti-Inflammatory Effect

ASA activity is primarily associated with dose-dependent COX-1 and COX-2 inhibition. The most common ones are (i) inhibition of nuclear factor kappa-light-chain-enhancer of activated B cells (NF-κB) pathway; (ii) endothelial nitric oxide (NO) release; and (iii) acting as a peroxisome proliferator-activated receptor gamma (PPARγ) agonist (Figure 2). Since these pathways were mostly investigated in vitro and in animal models and not always associated with myocardial ischemia, the results should be interpreted with caution, as they may not be directly applicable to patients undergoing CABG.

ASA improved alveolar bone defects healing in rats by impeding lipopolysaccharides (LPS)-induced macrophages via the inhibition of NF-κB pathway [30]. Moreover, ASA treatment improved the inflammatory response in cerebral infarction in mice thanks to downregulation of both TLR4 (toll-like receptor 4) and NF-κB expression, which led to endoplasmic reticulum (ER) stress inhibition [31]. Another promising target point for ASA is preeclampsia, which is known to depend on oxidative stress and inflammation via activation of NF-kB. By decreasing the translocation of NF-kB, low doses of ASA are used in preeclampsia prevention and treatment [32,33]. Also, ASA inhibits the growth and the metastasis of osteosarcoma through the NF-κB pathway [34].

Another alternative anti-inflammatory pathway of ASA is stimulation of direct endothelial NO release, independent of COX suppression, but presumably associated with endothelial nitric oxide synthase activation [35]. The NO-dependent effects of ASA include vasoprotection, gastric smooth muscle relaxation, and improvement of endothelial function, and they are directly associated with the reduction of cardiovascular risk. Moreover, ASA by indirectly activating alternate compensatory pathway including mainly heme oxygenase-1 and NO reduces the frequency of gastric injury. This gastric pathway works protectively, in contrast to the well-known ASA-dependent increased risk of disturbing gastrointestinal mucosal integrity [35,36].

The third possible pathway responsible for the anti-inflammatory effects of ASA is acting as a PPARγ agonist with subsequent downregulation of the WNT/β-catenin pathway. Upregulated WNT/β-catenin pathway participates in tumor growth, whereas PPARγ reduces inflammation, oxidative stress, cell proliferation, and invasion. NSAIDs such as ASA seem to work opposite on the two above mentioned pathways: downregulating WNT/β-catenin pathway and being a PPARγ agonist.

CABG is considered to be a trigger for the ‘cytokine storm’ phenomenon, which manifests in the increased concentrations of cytokines (e.g., interleukin 6 (IL-6), IL-8, tumor necrosis factor alpha) [37]. Taking into account that CABG is a potent activator of inflammation, ASA could be an important part in the postoperative therapy as an anti-inflammatory agent [4,5]. However, to achieve the ASA anti-inflammatory effect, it would be necessary to substantially increase the daily dose to at least 4 g per day.

## 4. Anticancerogenic Effect

There is increasing data demonstrating the role of ASA both in prevention and treatment in many cancer types. Cancer cells not only upregulate platelet production, but also generate extracellular vesicles (EVs) which stimulate platelet and leukocyte activation, leading to phosphatidylserine and tissue factor exposure on platelets and leukocytes and overall prothrombotic state in cancer patients [38].

The mechanisms of anticarcinogenic activity of ASA include: (i) inhibition of EV release; (ii) inhibition of the enzymatic activity of heparinase; and (iii) inhibition of epidermal growth factor receptor (EGFR) expression (Figure 2).

EVs are nanosized particles which are key players in intercellular communications [39] and have gathered a lot of attention as potential biomarkers of cancer development and progression [40,41]. For example, increased concentration of cancer-derived EVs was found in patients with many cancer types, including breast cancer [42], prostate cancer [43], or lung cancer [44]. Concurrently, ASA administration decreases both plasma concentrations of EVs and their procoagulant properties [45], suggesting that modulation of EV concentration and composition might be one of the mechanisms underlying ASA pleiotropic effects [46].

Heparanase is an extracellular matrix enzyme responsible for polymeric heparan sulphate degradation at the cell surface and in the extracellular matrix, which may be another key point for ASA application. By changing the constitution of extracellular matrix, heparanase promotes (i) cancer metastasis by assisting cancer cell migration and invasion and (ii) angiogenesis by releasing heparin-binding cytokines including hepatocyte growth factor (HGF), vascular endothelial growth factor (VEGF), and basic fibroblast growth factor (bFGF). By inhibiting heparanase activity, ASA was found to delay cancer progress (metastasis, angiogenesis) both in vitro and in vivo [47].

EGFR is a receptor commonly linked to epithelial malignancies [48]. It is upregulated in tumor microenvironment, resulting in tumor growth, invasion and metastasis [48]. ASA inhibits EGFR signaling, therefore exerting anticancerogenic effect in various cancer types. For example, ASA was shown to normalize EGFR expression and inhibit EGFR signaling both in human colorectal cancer (CRC) cells and ovarian cancer cells in vitro [32]. Consequently, ASA has gained a lot of attention as probably the most promising chemopreventive medication of CRC. Consequently, the United States Preventive Services Task Force recommended the use of low-dose ASA for the prevention of CRC in patients with specific cardiovascular profiles (aged 50–69 years with a 10% or greater 10-year cardiovascular risk, who are not at increased risk for bleeding, have a life expectancy of at least 10 years and are willing to take ASA for at least 10 years) [49]. Hence, ASA has become a part of the CRC prevention strategy [8].

In addition, ASA was shown to have a synergic effect with temozolomide, bevacizumab, and sunitinib in the glioblastoma therapy. The combined therapy of ASA and chemotherapeutics could improve the glioblastoma treatment efficacy [50]. Finally, a meta-analysis of four cohort studies and seven case-control studies including 653,828 participants with 12,637 incident cases showed that the increased dose of ASA (per two prescriptions per week increment) was related to a 5% relative risk reduction of head and neck cancers, compared to no prior use.

Since CAD and cancer are two prominent causes of death worldwide, the coexistence of both diseases is frequent. The prevalence of CAD in patients with different cancer types ranges from 43% in patients with lung cancer to 17% in patients with breast cancer [51]. Moreover, from the clinical experience the number of elderly patients undergoing CABG increases. Since age is a risk factor of cancer development, the above-mentioned group suffers often from both CAD and cancer simultaneously. Hence, a therapy which targets both disorders simultaneously seems promising. Taking into account that CAD and some cancer types have common risk factors including diabetes, obesity and smoking, the use of ASA as a complementary therapy in cancer patients might both prevent and/or delay cancer development, and limit the side effects of chemotherapeutics [6,7,8]. Thus, some experts have suggested to incorporate ASA in primary cancer prevention due to the reduction of cancer incidence and mortality after 3–5 years treatment with ASA [52].

## 5. Hypotensive Effect

ASA hypotensive effect has been the subject of many studies, and the majority of studies agreed on three outcomes: (i) low-dose ASA treatment is associated with a stronger decrease in blood pressure, compared to high-dose; (ii) evening ASA administration decreases blood pressure more effective than morning administration; and (iii) combination of ASA and statin reduces cardiovascular risk (Figure 2).

First, low-dose ASA treatment is associated with stronger decrease in blood pressure, in comparison to high-dose [53]. The underlying mechanism is based on the inhibition of COX-2 by high-dose ASA, but not by low-dose ASA. High-dose ASA leads to reduction of renal blood flow, glomerular filtration rate, sodium, and water excretion [53], which are the reasons for against using high doses of ASA in clinical practice. Another mechanism which seems to play a role in ASA hypotensive effect is a COX-independent pathway which revolves around proline-rich tyrosine kinase 2 (Pyk2). Pyk2 is a key player in RhoA/Rho activation, which is necessary for the vascular smooth muscle constriction. RhoA/rho is dysregulated in hypertension. Through the inhibition of the Pyk2-associated pathway in vascular smooth muscle cells, ASA and other salicylates may lower the blood pressure. This phenomenon is unique for ASA doses of higher concentrations than those required to inhibit COX [53].

Second, low doses of ASA may increase the efficacy of both antiplatelet and anti-hypertensive therapy, if administered in the evening. ASA administration in the evening increases its antiplatelet effect due to the release of young platelets from the bone marrow in the morning. The mechanism of ASA hypotensive effect, in turn, is possibly related to the increase in the nocturnal angiotensin II-dependent NO production [54]. Moreover, a greater decrease in blood pressure was noted in non-dipper patients who became dippers while taking ASA in the evening. The observed hypotensive effect in non-dippers is of great importance, because non-dippers are at higher risk of cardiovascular events.

Third, the combined therapy of ASA and statins is beneficial for cardiovascular risk reduction. The addition of ASA to statin therapy resulted in the decrease of SBP and DBP, in comparison to the control group treated with placebo [55,56]. Furthermore, the flow-mediated vasodilatation of brachial artery increased with aspirin-statin therapy, compared to placebo. However, it is inconclusive whether this effect was truly a result of the combined aspirin-statin therapy, or just the well-established impact of statins on endothelial functions [55].

About 24% of patients after CABG admitted for in-hospital cardiac rehabilitation presented with maximum systolic pressure of 200 mmHg and diastolic of 110 mmHg, caused most often by stress and withdrawal of antihypertensive medications used preoperatively [57]. The majority of patients have arterial hypertension prior to CABG, as it is a well-established risk factor for CAD development [58]. The most presumed mechanism underlying the development of CAD in hypertensive patients is the increase in pulsatile aortic wall stress, which enhances degradation of elastin, leading to arterial stiffness [59]. Arterial stiffness is strongly associated with vascular calcification, which is a major pathology underlying CAD development [60]. The synergy of action between ASA and statins demonstrates the complementary benefits of the combined therapy after CABG.

Altogether, ASA seems to have a hypotensive effect, especially when administered with statins. However, since the majority of previous studies were conducted in healthy volunteers or mildly hypertensive patients who were not treated with anti-hypertensive drugs before, there are yet no firm conclusions on the hypotensive effect of ASA and more studies are needed to clarify it [54].

## 6. Antiproliferative Effect

It has been established that ASA exerts antiproliferative effect on VSMCs in patients after CABG [61]. So far, several mechanisms underlying this specific effect of ASA have been discovered. They are mainly related to (i) transforming growth factor β1 (TGF-β1) and (ii) platelet-derived growth factor (PDGF) (Figure 2).

Primarily, ASA increases the secretion of TGF-β1. TGF-β1 directly inhibits VSMCs proliferation by inducing G0/G1 phase cell cycle arrest [62]. Additionally, TGF-β1 reduces the production of matrix metalloelastase (MMP-12) in macrophages [63]. MMP-12 is one of metalloproteinases—enzymes associated with instability and rupture of atherosclerotic plaque. By increasing TGF-β1 secretion, ASA therapy might promote atherosclerotic plaque stability.

Second, ASA suppresses PDGF release from thrombocytes [64]. PDGF is one of the most important growth factors for VSMCs. Its excessive expression contributes to the development of atherosclerosis or organ fibrosis, but also to the late graft occlusion and late restenosis in patients after CABG [65,66].

One in vitro study evaluated the effect of ASA on PDGF-treated VSMCs and on retinoblastoma protein hyperphosphorylation, which is known to regulate cell cycle progression. This study found that ASA arrests the cell cycle also at the G1/S phase. Therefore, the high-dose of ASA treatment may be beneficial in treatment of vascular proliferative disorders [67].

The main factors which determine graft failure after CABG include endothelial damage, thrombosis and VSMCs proliferation. VSMC proliferation is associated with atherosclerotic plaque formation and neointimal hyperplasia, which lead to graft stenosis and finally to (mainly late) graft occlusion [9]. ASA therapy seems to prevent excessive and dysfunctional VSMC proliferation, therefore decreasing the risk of late graft failure and further suppressing atherosclerosis progression [9].

## 7. Influence on Bone Mineral Density

Osteoporosis is a metabolic bone disease that mainly affects elderly, postmenopausal women. It is associated with low bone mineral density (BMD) which can be measured by dual-energy X-ray absorptiometry scan. Osteoporosis leads to pathologic fractures of the hip, spine, and wrist and is a significant cause of disability and mortality among elderly patients. Osteoporosis develops due to the imbalance between bone formation and resorption, with the latter being predominant. It has been established that ASA affects bone remodeling. PGE2 stimulates the proliferation of osteoclast precursors and their differentiation and transformation to mature osteoclasts [68]. Furthermore, PGE2 stimulates bone resorption and may contribute to increased bone loss [69]. By inhibiting COX-2 activity, ASA reduces PGE2 production. Furthermore, ASA promotes the survival of bone marrow mesenchymal stem cells [70], the progenitors of osteoblasts. In this manner, ASA supports bone formation and decreases bone resorption.

Prolonged ASA intake was associated with higher total hip and lumbar BMD in women [71,72,73] and men [71,72,74] and with lower fracture risk in observational studies [72]. For example, ASA users between the age of 70–79 were found to have higher BMD, compared with non-users [71]. In a multicenter study which included 7786 women aged ≥65 years, the hip and lumbar BMD was higher in patients who used ASA for more than a year, compared to non-users [73]. However, more studies should be conducted to confirm these findings, as the heretofore conducted studies had methodological differences including different study groups, different ASA dosages, and questionable patient compliance [75].

Patients undergoing CABG are more likely to develop osteoporosis due to several risk factors, such as elderly age or immobilization after surgery. A study comprising 26 patients revealed considerable, progressive bone mineral density decrease during the first year after CABG [76]. Despite the small sample size, this study implies that patients undergoing CABG may be at higher risk of osteoporosis. Fragility and osteoporotic fractures significantly worsen the quality of life, diminish physical activity and may disturb rehabilitation after CABG. Hence, ASA therapy, even in low doses (<150 mg) may bring additional benefits of BMD increase and reduced fracture risk.

## 8. Neuroprotective Effect

Thanks to its potential neuroprotective effect, ASA found its place in the neuropsychiatric field in the treatment of Alzheimer’s disease, schizophrenia, bipolar disease, and depression. The precise mechanism underlying ASA neuroprotective effect has not been well established yet. However, studies show few promising possibilities related to (i) PPARα, (ii) brain-derived neurotrophic factor (BDNF), (iii) COX-1 and COX-2, and (iv) glutamate excitotoxicity (Figure 2).

ASA binding to PPARα was found to up-regulate the hippocampal plasticity [77]. Moreover, ASA was found to enhance expression of BDNF messenger RNA in the neurons of the hippocampus in mice. BDNF is a neurotrophin promoting cell survival and synaptic plasticity which plays a key role in pathogenesis of different neurodegenerative diseases, such as Alzheimer’s disease [78]. Accordingly, low-dose ASA was observed to improve spatial learning and memory in animal model of Alzheimer’s disease [77].

Another pathway underlying ASA neuroprotective function may involve COX-2 modification and COX-1 irreversible inhibition, which leads to suppression of inflammatory reaction [79]. As schizophrenia has an inflammatory background, ASA is recommended also in the treatment of the first episode of schizophrenia [80]. However, in schizophrenia, high doses of ASA are needed because of difficulties to cross the blood brain barrier [79].

ASA anti-inflammatory activity appears to improve symptoms of mood disorders, such as depression or bipolar relapse [81]. Another animal study demonstrated ASA positive impact on oligodendrocyte precursor cell (OPCs) proliferation and differentiation after white matter lesion, using low and high doses of ASA, respectively [82]. It was shown that low doses of ASA (25 mg/kg) increase the amount of OPCs, while relatively high doses of ASA (100–200 mg/kg) elevate the number of oligodendrocytes and improve myelin thickness after white matter lesions [82]. Thus, both doses seemed to have neuroprotective effects.

A case report described a woman with electroencephalogram abnormalities in the left temporal-occipital area reversed and symptoms relieved after low-dose ASA therapy alone. A potential mechanism underlying the neuroprotective abilities of ASA seems to be related to its activity against glutamate excitotoxicity. However, this hypothesis is based on case reports and more research should be carried out to prove a direct link between ASA and neurotransmitter dysfunction [83].

CABG is a procedure performed in the elderly patients who are known to be at higher risk of atherosclerotic vascular changes. Moreover, atherosclerosis is an established risk factor of Alzheimer’s disease. In addition, especially elderly patients who underwent CABG are susceptible to cognitive impairment and dementia [10]. Therefore, the neuroprotective features of ASA related to BDNF, PPARα, and glutamate excitotoxicity may complement the standard therapy for patients suffering from both cognitive impairment/dementia and atherosclerosis. Altogether, the framework of ASA as a neuroprotective agent is broad, but furthermore, more specific research is essential to draw firm conclusions.

## 9. COVID-19 and ASA

In the times of global pandemic, the application of various drugs in the treatment and prophylaxis of COVID-19 and the following complications is an important research topic. Accumulating data prove the relationship between COVID-19 and thrombosis [84]. The clinical and autopsy studies indicate that the risks of microvascular thrombosis, venous thromboembolism, and ischemic stroke are higher in COVID-19 patients [85,86]. Also, it was found that platelets have high affinity receptors for SARS-CoV-2, suggesting that platelets may participate in the development of COVID-19 [85]. Thus, the application of antiplatelet drugs in the prophylaxis and treatment of COVID-19 and related complications might be a viable treatment strategy. Currently, the clinical data regarding the efficacy of ASA to improve outcomes in COVID-19 patients remains ambiguous. CAD patients treated with ASA (75–150 mg per day) prior to COVID-19-related hospitalization have comparable in-hospital mortality as patients not receiving ASA (21.2% vs. 22.1%) [87]. On the other hand, when comparing in-hospital ASA intake to no antiplatelet therapy, a decrease in the rate of in-hospital death in the ASA group was found [88]. In addition, ASA in-hospital administration was associated with lower need of mechanical ventilation, in comparison to the non-ASA group [89]. Altogether, more studies are needed to obtain reliable conclusions regarding the effect of ASA use in the prophylaxis and treatment of COVID-19 and related complications.

## 10. Conclusions and Future Directions

In spite of ASA’s long medical history, novel possible applications for ASA therapy still arise. Recent data demonstrates a variety of new targets for ASA beyond platelet inhibition, including cancer, hypertension, vascular cell muscle proliferation, osteoporosis, and neurological impairment. Many of these disorders have a higher prevalence in patients undergoing CABG, demonstrating the potential benefits of ASA beyond platelet inhibition. The pathophysiological mechanisms underlying these disorders after CABG include peripheral and central atherosclerosis, thrombosis, hypertension, and risk of fractures. They are often based on chronic inflammatory state associated with such pathways as WNT/β-catenin, NF-κB, PPARα, PPARγ, and different growth factors (e.g., BDNF, TGF-β1) which are affected by ASA. Some of the proposed mechanisms remain innovative and not evident yet, as they were observed in vitro and in animal models only. Many of the above-described mechanisms were not confirmed in the CABG patients and thus, they should be interpreted with caution. The risks of adverse effects of ASA (e.g., bleeding) should be taken into account, especially when the higher dose therapy of ASA is considered.

Considering ASA pleiotropic effect, a hypothesis arises that the intensification of ASA therapy might improve outcomes after CABG. The American Heart Association guidelines recommend the use of 81–325 mg ASA daily to reduce the risk of graft occlusion and adverse cardiac events [1]. One meta-analysis implied that the intermediate ASA dose (300–325 mg) may decrease the rate of reduce graft occlusion, compared to the low dose (75–160 mg) regimes within the first year after CABG [90]. There is a need for further clinical studies to establish whether the intermediate dose is superior to low dose in patients after CABG, taking into account both the graft patency and other beneficial effects; including hypotensive or neuroprotective effects, as well as adverse effects such as the risk of bleeding, gastric complaints, and reduction of glomerular filtration rate.

It is important to mention another antiplatelet drug with pleiotropic effects—ticagrelor, which could be combined with ASA to maximize the treatment benefits [91]. In the Dual Ticagrelor Plus Aspirin Antiplatelet Strategy After Coronary Artery Bypass Grafting (DACAB) study, 500 patients were randomized to receive ticagrelor (90 mg twice daily) + ASA (100 mg once daily), ticagrelor (90 mg twice daily), or ASA (100 mg once daily) within 24 h post-CABG. Although the combination of ticagrelor and ASA was superior to aspirin alone in maintaining the saphenous vein graft patency for up to 1 year after elective CABG, there were no differences between ticagrelor and ASA. Although the rate of cardiovascular events and bleeding events was low, the combination of ticagrelor and ASA numerically decreased the risk of the major cardiovascular events after CABG (cardiovascular death, myocardial infarction, stroke), but also increased the risk of bleeding (CABG-related, non-CABG-related, and major bleeding events) [92], compared to ASA. The main results of DACAB study are presented in Figure 3. Altogether, the combinations of ticagrelor and ASA might improve graft patency after CABG, but further studies are needed to assess the bleeding risk.

In another population—patients undergoing transcatheter aortic valve implantation (TAVI), those treated with ASA monotherapy for 3 months after TAVI has lower incidence of a composite end-point of bleeding and thrombotic events at 1 year, compared to ASA and clopidogrel [93]. Hence, ASA monotherapy seems to be associated with decreased bleeding risk, compared to dual antiplatelet therapy. It could be hypothesized that increasing the ASA dose might also be associated with reduced rate major cardiovascular events. However, in absence if evidence-based data, this hypothesis remains a speculation.

Until now, only a few studies have compared the bleeding risk with different ASA doses. For example, a review of 39 studies conducted in the field of gastroenterology proved that the risk for upper gastrointestinal bleeding was comparable, regardless of the ASA dose and treatment duration [94]. Hence, it is crucial to investigate whether higher doses of ASA indeed significantly increase the bleeding risk, and if so, whether it is higher than the bleeding risk during dual antiplatelet therapy.

Altogether, individualization of the postoperative dose of ASA after CABG based on the patients’ individual risk of graft occlusion and bleeding is worth consideration. At least in some patients, the possible benefits of ASA pleiotropism might outweigh the risk of bleeding. However, these groups of patients still remain to be identified. Moreover, although the results of pre-clinical studies are very promising, the evidence-based data from randomized controlled trials are lacking. Further research is required to confirm the pleiotropic effects of ASA in the clinical setting.

## Figures and Tables

**Figure 1 jcm-10-02317-f001:**
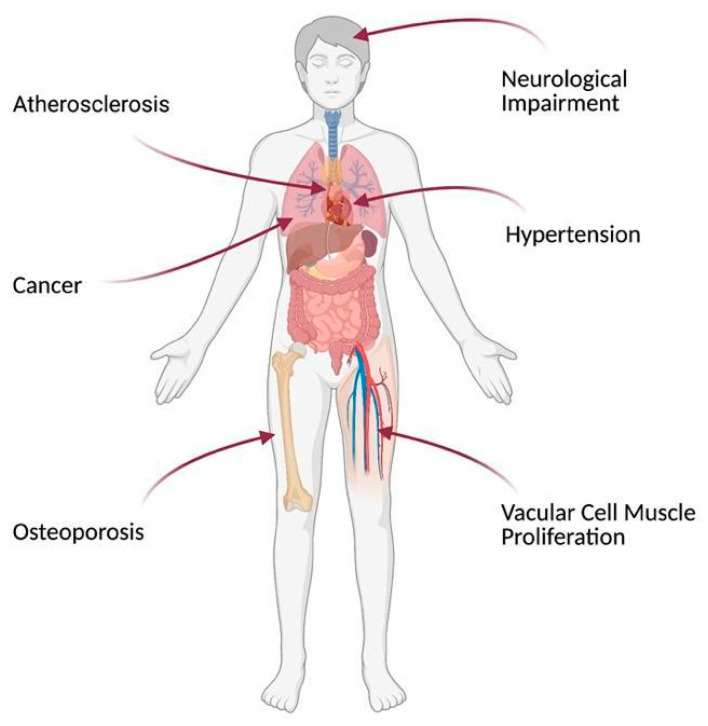
Possible targets for pleiotropic effects of acetylsalicylic acid in patients after coronary artery bypass grafting Figure created with BioRender.com.

**Figure 2 jcm-10-02317-f002:**
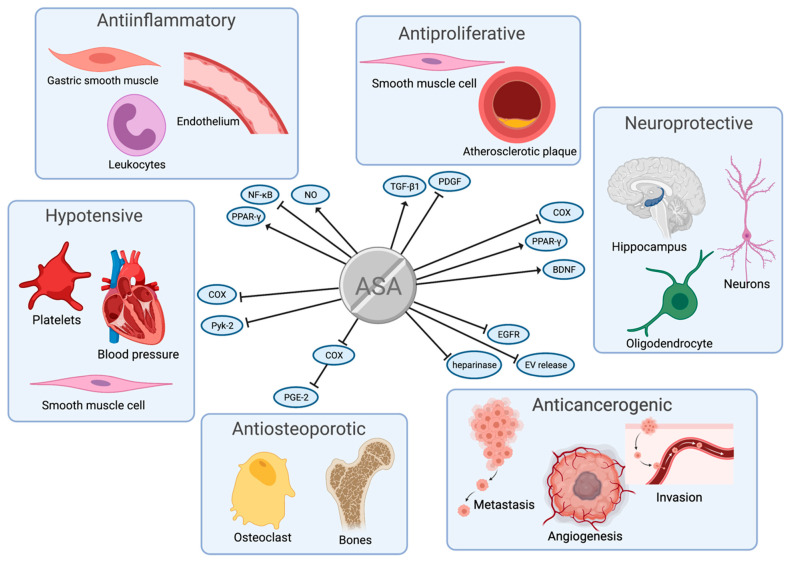
Pleiotropic effects of acetylsalicylic acid (ASA) discussed in this review. For details regarding the mechanisms, please see the main text. BDNF—brain-derived neurotrophic factor; COX—cyclooxygenase; EGFR—epidermal growth factor receptor; EV—extracellular vesicles; NF-κB—nuclear factor kappa-light-chain-enhancer of activated B cells; NO—nitric oxide; PDGF—platelet-derived growth factor; PGE-2—prostaglandin E-2; PPARγ—peroxisome proliferator-activated receptor gamma; Pyk-2—proline-rich tyrosine kinase 2; TGF-β1- transforming growth factor β1. Figure created with BioRender.com.

**Figure 3 jcm-10-02317-f003:**
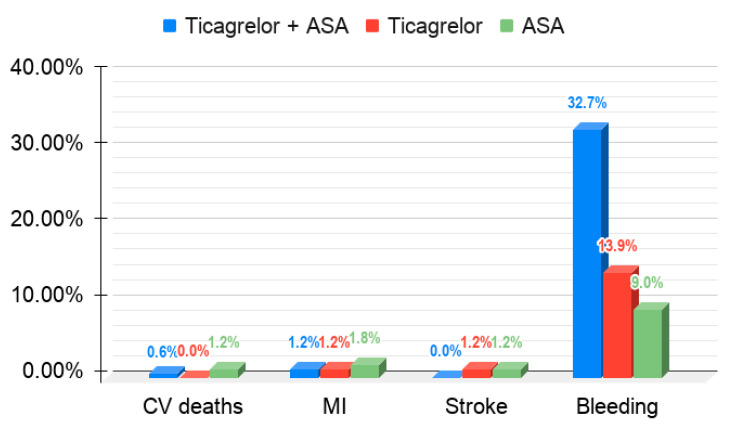
Postoperative complications after CABG with different treatment regimens. CV deaths—cardiovascular deaths; MI—myocardial infarction.

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
