# Peer review of "Pleiotropic Effects of Acetylsalicylic Acid after Coronary Artery Bypass Grafting—Beyond Platelet Inhibition"

_jcm, 2021, doi:10.3390/jcm10112317_

Round 1

Reviewer 1 Report

Dear Authors

The presented article is an overview of the different actions of acetylsalicilic acid  on possible pathophysiologic mechanisms involved in atherosclerosis, smooth muscle proliferation , antiinflammmatory effects and more distantly its presumed  hypotensive and cognitive function  and beneficial effects in oncological and bone regulation  aspects

There is a lot of positive actions of this very largely used old drug presented in the article,  which are not fully balanced with possible negative aspects of its extended use, especially in higher doses. Many years ago, acetylsalicilic acid was used in higher doses in rheumatology , infections and other clinical, situations, however those doses were poorly tolerated by a large number of patients that was one of the main reasons to decrease the dosage in the current era.

It should  also be noted that it was not approved as prophylaxis of cardiovascular disease for healthy people due to unfavorable balance of positive and negative effects in this population.

My main critical remarks are:

  • The proportion of risk to benefit of large use of acetylosalicic acid in your text in not well balanced
  • The use of acetylosalicilic acid in different clinical settings is presented with only a few connotations to the dosage.
  • The results of many small studies are presented and mixed with larger randomized studies, which can mislead the reader in the true clinical significance of those studies.
  • The proposition already seen in abstract lines 38 and 39 “ Since the plethora of ASA pleiotropic effects seem to prevail the risk of bleeding, it could be considered to increase the intensify the treatment  with ASA after CABG.”  In my opinion is “ one bridge too far”. Very sparse clinical data justify such a far-reaching conclusion. The “plethora of of pleiotropic effects” are only presumed and not proven in this clinical setting
  • There is no differentiation between early and late use of large doses of aspirin in CABG patients – risk of gastrointestinal bleeding, possible secondary post-op bleeding
  • This article is an interesting overview of the different positive actions of acetylsalicylic acid, but justification of the use of higher doses is extremely speculative and not justified by my opinion in EBM data

Author Response

Dear Reviewer,

we are thankful for the time and effort that you spent to provide in-depth review of our review. We corrected our manuscript according to your suggestions. Our response and

corrections are listed below. Please see the attachment.

Reviewer 2:

The presented article is an overview of the different actions of acetylsalicylic acid on possible pathophysiologic mechanisms involved in atherosclerosis, smooth muscle proliferation , anti-inflammatory effects and more distantly its presumed  hypotensive and cognitive function  and beneficial effects in oncological and bone regulation  aspects

There is a lot of positive actions of this very largely used old drug presented in the article, which are not fully balanced with possible negative aspects of its extended use, especially in higher doses. Many years ago, acetylsalicylic acid was used in higher doses in rheumatology, infections and other clinical, situations, however those doses were poorly tolerated by a large number of patients that was one of the main reasons to decrease the dosage in the current era. It should  also be noted that it was not approved as prophylaxis of cardiovascular disease for healthy people due to unfavourable balance of positive and negative effects in this population.

My main critical remarks are:

  1. The proportion of risk to benefit of large use of acetylosalicic acid in your text in not well balanced

We thank the reviewer for this remark. We added a paragraph considering the major adverse effect of ASA - bleeding in our introduction.

  1. The use of acetylosalicilic acid in different clinical settings is presented with only a few connotations to the dosage.

We thank the reviewer for this remark. We wanted to focus on the overall actions of ASA. We mentioned the dosages only when we found it important for diversifying the effect of ASA in the clinical situation.

  1. The results of many small studies are presented and mixed with larger randomized studies, which can mislead the reader in the true clinical significance of those studies.

We thank the reviewer for this remark. In most of the studies we mentioned we added the number of study participants. We added information to the study sample size or its character (animal, in vitro) where it was previously not provided.

  1. The proposition already seen in abstract lines 38 and 39 “Since the plethora of ASA pleiotropic effects seem to prevail the risk of bleeding, it could be considered to increase the intensify the treatment  with ASA after CABG.”  In my opinion is “ one bridge too far”. Very sparse clinical data justify such a far-reaching conclusion. The “plethora of of pleiotropic effects” are only presumed and not proven in this clinical setting.

We agree with the Reviewer that the suggestion was far-fetched. We revised the whole article in a more cautions manner.

  1. There is no differentiation between early and late use of large doses of aspirin in CABG patients – risk of gastrointestinal bleeding, possible secondary post-op bleeding

We thank the reviewer for this remark. We added a paragraph in our Introduction considering bleeding as ASA adverse effect directly after CABG and included the information about the follow-up time, where possible.

  1. This article is an interesting overview of the different positive actions of acetylsalicylic acid, but justification of the use of higher doses is extremely speculative and not justified by my opinion in EBM data.

We thank the reviewer for this suggestion. We underlined in the conclusions that the EBM data are lacking and there is a need for more research to ascertain the presence in the clinical setting the benefits which were found in cell cultures or animal models.

Altogether, we are grateful for the in-depth revision of our manuscript and we hope that it will be considered for publication.

On behalf of all Authors,

Sincerely,

Dominika Siwik,

Magdalena Gajewska,

Aleksandra Gasecka

Reviewer 2 Report

This is an interesting review on the pleiotropic biological activities of acetyl-salicylic acid (ASA) in medical applications. It is a well-redacted and well-documented article. The major comment is that this review is too much like a pleading to extend the ASA use in many clinical contexts, especially in prevention of arterial thrombosis, control of inflammation, oesteoporosis and cancer. This is indeed a valuable presentation, but it should be useful to also show the adverse effects of ASA and to discuss the risk-benefit ratio in the various clinical contexts. The major issue with ASA intake is bleeding. Only figure 3 shows the bleeding risk, which is increased when ASA is associated with Ticagrelor as compared to Ticagrelor or ASA alone. In the text bleeding is only evoked at line 479. Authors should balance the beneficial effects of ASA, which are obvious, with the cautions or the requirements for patients' survey and the monitoring to implement for avoiding the bleeding risk. ASA counterindications could also concern certain patients more exposed to hemogrrhages, or elderly persons already treated with anticoagulants.

I have only few minor comments:
Line 160:  the action of PGH2 on peroxidase and Cox is not clearly described. The sentence is confusing and needs revision.
Line 166: it should be "produce" and not "reproduce".
Line 181 this sentence is redundant with the former paragraph, and provides a confusing information as the ASA doses required for Cox 1 and Cox 2 inhibition differ.
Lines 304-305: the presentation needs to be corrected for the extra spaces.

Author Response

Dear Reviewer,

we are thankful for the time and effort that you spent to provide in-depth review of our review. We corrected our manuscript according to your suggestions. Our response and

corrections are listed below.

Reviewer 1:

This is an interesting review on the pleiotropic biological activities of acetyl-salicylic acid (ASA) in medical applications. It is a well-redacted and well-documented article. The major comment is that this review is too much like a pleading to extend the ASA use in many clinical contexts, especially in prevention of arterial thrombosis, control of inflammation, oesteoporosis and cancer. This is indeed a valuable presentation, but it should be useful to also show the adverse effects of ASA and to discuss the risk-benefit ratio in the various clinical contexts. The major issue with ASA intake is bleeding. Only figure 3 shows the bleeding risk, which is increased when ASA is associated with Ticagrelor as compared to Ticagrelor or ASA alone. In the text bleeding is only evoked at line 479. Authors should balance the beneficial effects of ASA, which are obvious, with the cautions or the requirements for patients' survey and the monitoring to implement for avoiding the bleeding risk. ASA counterindications could also concern certain patients more exposed to hemogrrhages, or elderly persons already treated with anticoagulants.

We thank the Reviewer for appreciating our Review and for the suggestions to extend the part about bleeding. We added a paragraph considering the major adverse effect of ASA - bleeding in the introduction (marked red).

I have only few minor comments:
1)  the action of PGH2 on peroxidase and Cox is not clearly described. The sentence is confusing and needs revision.

2) it should be "produce" and not "reproduce".

3) this sentence is redundant with the former paragraph, and provides a confusing information as the ASA doses required for Cox 1 and Cox 2 inhibition differ.

4) the presentation needs to be corrected for the extra spaces.

We revised and corrected the above-mentioned parts according to the Reviewer’s suggestion. Please see the attachment.

Altogether, we are grateful for the in-depth revision of our manuscript and we hope that it will be considered for publication.

On behalf of all Authors,

Sincerely,

Dominika Siwik,

Magdalena Gajewska,

Aleksandra Gasecka

Round 2

Reviewer 2 Report

Many thanks to the authors for their rapid revision of the article and positively considering the changes suggested.

This article looks now better balanced between benefits and risks of acetylsalicylic acid use, and information provided shows a higher objectivity level. 

The comments have been correctly addressed and I have no additional request.